# The efficacy of pediatric elbow radiographic guidance in diagnosis of lateral humeral condyle fracture

Satetha Vasaruchapong[1], Patarawan Woratanarat[2], Tanyaporn Patathong[2], Thira Woratanarat[3], Supaneewan Jaovisidha[4], Chanika Angsanunsukh[2]*

1 Chakri Naruebodindra Medical Institute, Faculty of Medicine Ramathibodi Hospital, Mahidol University, Salaya, Thailand, 2 Department of Orthopedics, Faculty of Medicine Ramathibodi Hospital, Mahidol University, Salaya, Thailand, 3 Department of Preventive and Social Medicine, Faculty of Medicine, Chulalongkorn University, Bangkok, Thailand, 4 Department of Diagnostic and Therapeutic Radiology, Faculty of Medicine Ramathibodi Hospital, Mahidol University, Salaya, Thailand

* achanika@gmail.com

**Data Availability Statement:** All relevant data are within the manuscript and its Supporting Information files.

## Abstract

Although lateral humeral condyle fracture is common, the incidence of missed diagnosis is very high. Delayed and missed diagnosis led to significant morbidities and loss of functions. We designed a pediatric elbow radiographic guidance aiming to improve the accuracy of diagnosis. This study was aimed to evaluate the efficacy of the radiographic guidance for the diagnosis of lateral condyle fracture. A cross-sectional study was conducted after defining the essential parameters as a guidance for assessing the pediatric elbow radiographs. We included medical students, emergency medicine, orthopedic, and radiology residents and fellows into this study. A questionnaire was used to evaluate the efficacy of the guidance. All participants underwent a pretest evaluation, followed by studying the guidance, and then finished a posttest evaluation. Baseline characteristics, diagnostic scores, and parameter evaluation scores were collected. The pretest and posttest scores were analyzed using paired t-test. Association between baseline characteristics and diagnostic scores were analyzed using multiple regression analysis. We included 177 participants. Average diagnostic score was significantly increased after using the guidance, from 12.2 ± 1.9 to 13.0 ± 1.7 (p < 0.0001). Medical students showed the most improvement, from 11.9 ± 1.9 to 13.1 ± 1.3 (p <0.001). All means of essential parameter evaluation scores were significantly improved in overall participants.The pediatric elbow radiographic guidance is useful for evaluation and diagnosis of lateral condyle fracture, especially for young physicians and trainees. Therefore, this should be recommended in routine medical education and general practice.

## 1. Introduction

Lateral condyle fracture is one of the most common elbow fractures in children [1–4]. The incidence is 1.6 per 10,000 children per year, which accounts for 10–20% of pediatric elbow

**Funding:** The author(s) received no specific funding for this work.

**Competing interests:** The authors have declared that no competing interests exist.

fractures [5, 6]. Even though lateral condyle fracture is very common, the diagnosis has been reported to be missed by clinicians. For instances, Shrader MW et al. reported that 60% of lateral condyle fracture were missed by general practitioners and emergency physicians [7].

Missed diagnosis of lateral condyle fracture can lead to non-union, malunion, limited range of motion, functional disturbance, varus/ valgus deformities and nerve palsy [5, 6]. Some of these consequences result in permanent deformities and morbidity [8]. Therefore, accurate diagnosis and appropriate treatments are crucial for preventing the complications. To diagnose lateral condyle fracture, the physicians need to gather data from history taking, physical examination, and radiographic evaluation. Nevertheless, history taking and physical examination in young patients are challenging and sometime unreliable. Therefore, standard radiographs of the elbow are essential for the diagnosis [9]. Evaluation of pediatric elbow radiographs is difficult especially for trainees and general practitioners. Many essential radiographic parameters include the posterior fat pad sign, anterior humeral line, radiocapitellar line, Baumann's angle, and fracture line at lateral condyle on both anterior-posterior (AP) and lateral radiographs. Moreover, the proper position of the elbow and quality of the radiographs are important [10–14].

According to data from the medical council of Thailand, there are only 2,500 orthopedic surgeons in the country taking care of more than 69 million population [15, 16]. The proportion of orthopedic surgeon per population is 1 per 27,600. Additionally, a large number of them are working only in big city areas. Therefore, the physicians who are early responsible for diagnosis of lateral condylar fracture are general practitioners due to limited numbers of orthopedic surgeon, but accurate diagnosis may be difficult and requires professional experiences. The authors initiated a radiographic guidance aiming to help the physicians to improve accuracy of diagnosis of lateral condyle fracture in children.

The purpose of this study was to evaluate the efficacy of the pediatric elbow radiographic guidance in diagnosis of lateral condyle fracture among inexperienced physicians and trainees, including medical students, emergency medicine residents, orthopedic residents and fellows and radiology residents and fellows.

## 2. Materials and methods

### 2.1 Study design

A cross-sectional study was conducted after receiving approval from the Ethics Review Committee of Ramathibodi Hospital, affiliated with Mahidol University (ID 08–58–08). Each participant provided informed written consent before being included in the study. The study was conducted from September 2015 to September 2016.

### 2.2 Subject and methods

We recruited medical students who have already finished orthopedic rotation, emergency medicine residents, orthopedic residents and fellows and radiology residents and fellows.

The literature reviews together with referencing the standard textbooks was done to identify the essential radiographical parameters for the diagnosis of elbow fractures in children. We then developed a pediatric elbow radiographic guidance explaining the definitions, providing the examples of normal and abnormal radiographs for each essential parameter. The steps for evaluating the radiographs were orderly exhibited. Starting with the first step is to assess the quality of the radiographs in both AP and lateral views, then look for the obvious fracture. (Fig 1) If the fracture could not be clearly identified, the next step is to look for other five essential parameters to evaluate the radiographs including posterior fat pad sign, anterior humeral line, radiocapitellar line, fracture line in both AP and lateral views, and soft tissue swelling.

## Radiographic evaluation guideline of pediatric elbow injury

1. Assess clinical symptoms: swelling, bruising, tenderness, and Isosceles triangle.

2. Evaluate the quality of radiographs for both exposure and position.

3. Considering both anteroposterior and lateral view

| AP view | Lateral view |
|---|---|
| - Soft tissue | - Fat pad sign |
| - Radiocapitellar line | - Anterior humeral line |
| - Fracture line | - Radiocapitellar line |
| | - Fracture line |

## Normal radiographs of a child's elbow

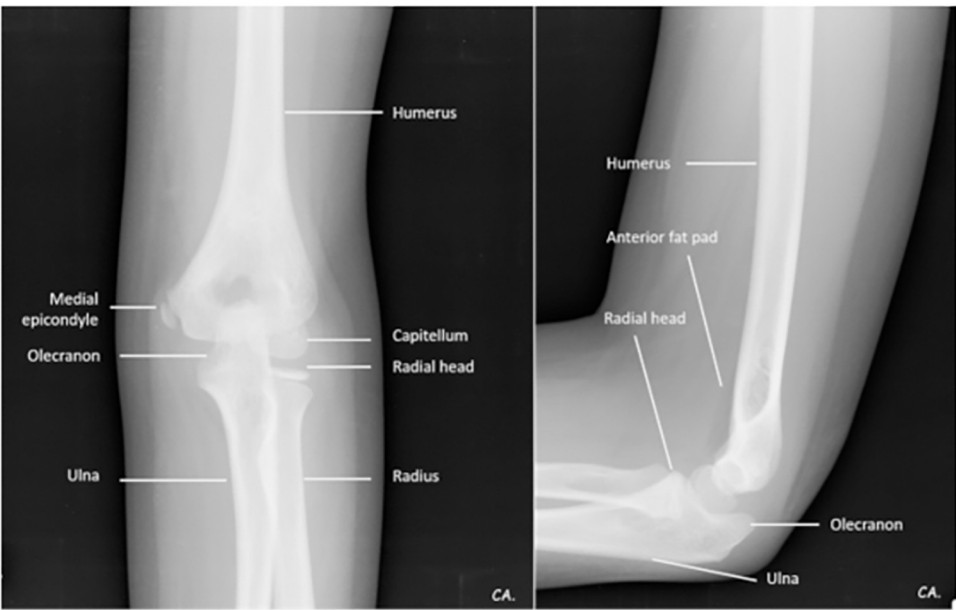

**Fig 1. Pediatric elbow radiographic evaluation guidance (1st page).**

(Figs 2–5) These steps of five parameter evaluation were called "PARFS" (Posterior fat pad sign, Anterior humeral line, Radiocapitellar line, Fracture line, and Soft tissue swelling, respectively).

A questionnaire was created for pre-test and post-test evaluations in order to assess the efficacy of the guidance. (Figs 6 and 7) The questionnaire consisted of 16 radiographs, including 5 normal cases, 2 supracondylar fractures, and 9 cases of lateral condyle fractures of various degrees. Details of radiographs such as patients' identification, serial numbers, data and time were blinded. All radiographs had been independently evaluated and confirmed for diagnosis by two experienced pediatric orthopedic surgeons. Each correct radiographic diagnosis was scored as 1 point, with a total possible score of 16 for the 16 radiographs.

After obtaining informed consent from the participants, we collected baseline characteristics, including age, gender, work experience, number of case experiences, orthopedic grade in

**Lateral view**

1. Posterior fat pad sign is a radiolucent crescent sign located in the olecranon fossa on a true lateral view.

   Positive posterior fat pad sign results from distension of the fat pad indicating effusion or bleeding.

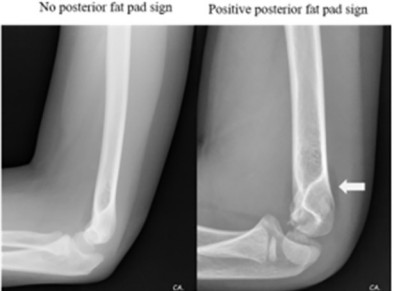

2. Anterior humeral line is a line drawn along anterior cortex of distal humerus towards the elbow joint.

   In a normal elbow radiograph, anterior humeral line should pass through middle third of the capitellum.

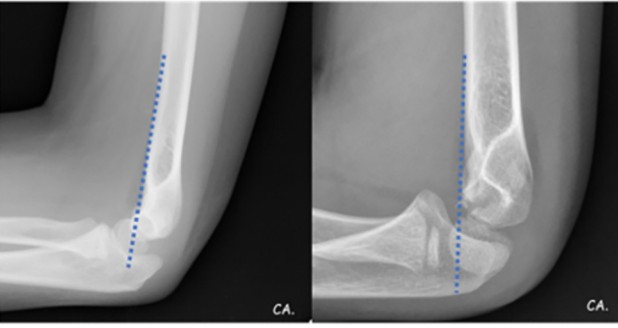

**Fig 2. Pediatric elbow radiographic evaluation guidance (2nd page).**

medical school, and experience in orthopedic internship. Participants were then asked to evaluate the questionnaire and record their evaluations on pre-test answer sheets. Subsequently, each participant was provided with the pediatric elbow radiographic guidance for 10 minutes of self-study. After finishing the study, they were asked to evaluate the questionnaire containing 16 radiographs in different orders from pre-test, and answered in post-test answer sheet. Baseline characteristics of participants, their radiographic evaluations, and diagnosis for each case from both pre-test and post-test were collected. We collected data in each department and included all participants who were willing to participate the study. There was no responders because all participants answer the questionnaire at the same time in department meeting room.

### 2.3 Statistical analysis

The sample size was calculated based on a pilot study of each group. The alpha error was 0.05 and power of the study was 0.8. Mean difference of pre-test and post-test diagnostic scores and standard deviations were 2.6 ± 2.39, 1.0 ± 1.92, 1.0 ± 1.09, and 0.79 ± 1.44 in medical students, emergency medicine residents, orthopedic residents and fellows and radiology residents and fellows, respectively. Sample size calculation were 9 for medical students, 31 for emergency

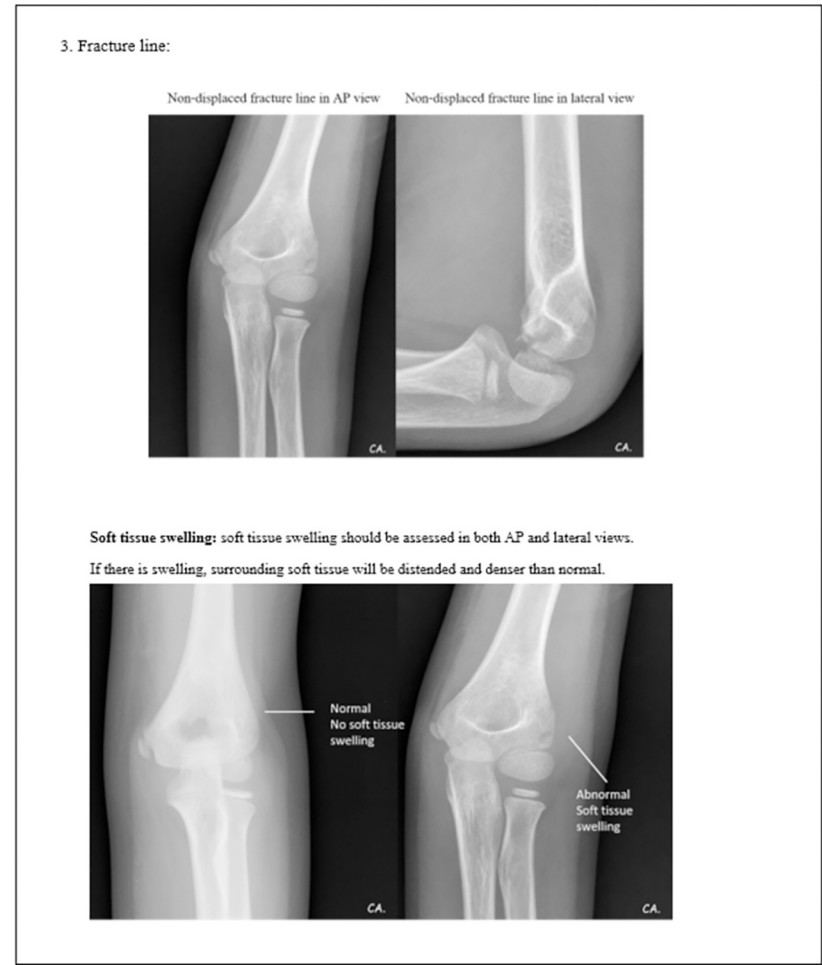

**Fig 3. Pediatric elbow radiographic evaluation guidance (3rd page).**

medicine residents, 11 for orthopedic residents and fellows and 28 for radiology residents and fellows. Pre-test and post-test parameter evaluation scores and diagnostic scores were analyzed using paired t-test. The association between baseline characteristics and diagnostic scores improvement was tested using multiple regression analysis. The statistical analysis was done using STATA 13.0 (StataCorp, College Station, Texas, USA).

## 3. Results

One hundred and seventy-seven participants were enrolled in this study. There were 60 of medical students, 32 of emergency medicine residents, 57 of orthopedic residents and fellows, and 28 radiology residents and fellows. Baseline characteristics were significantly different in each group which represent the divergence of each group. Baseline characteristics were presented in **Table 1**.

For the overall participants, mean of diagnostic scores was significantly improved from 12.4 ± 2.0 to 13.4 ± 1.4 after using the guidance (p = 0.003). Medical student was the most improved group, mean diagnostic score was improved from 11.9 ± 1.9 to 13.1 ± 1.3 (p <0.001), followed by emergency medicine (10.9 ± 1.8 to 11.8 ± 2.2, p < 0.001) and orthopedic residents and fellows (12.9 ± 1.6 to 13.7 ± 1.3, p = 0.002). For radiology residents and fellows,

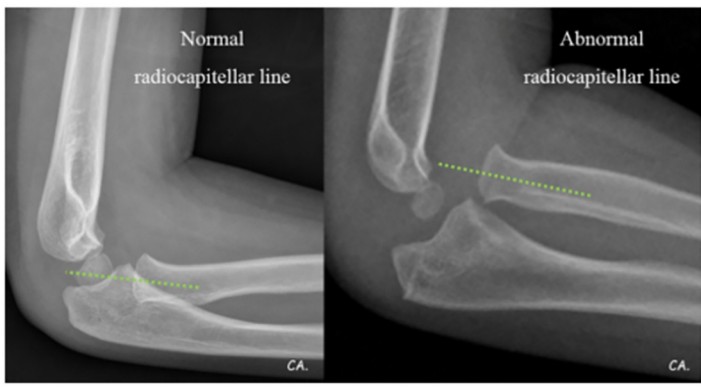

**Fracture line:** fracture line can be identified as a radiolucent line at lateral condyle. In non-displaced fracture, we should focus on non-parallel or irregular growth plate and displaced metaphyseal fragment.

**Radiocapitellar line:** radiocapitellar line is a line drawn along axis of proximal radius in AP and lateral views. The line should pass through middle third of the capitellum. Abnormal Radiocapitellar line indicates abnormality of radial neck, radial head, or capitellum.

**Fig 4. Pediatric elbow radiographic evaluation guidance (4th page).**

the mean of diagnostic scores was not significantly improved (12.5 ± 1.6 to 12.8 ± 1.5, p = 0.558) as shown in **Table 2**.

Means of parameter evaluation scores were significantly improved for all four parameters in overall participants. The mean of posterior fat pad sign evaluation scores was increased from 11.2 ± 2.0 to 12.1 ± 1.7 (p < 0.001). Mean of anterior humeral line evaluation scores were improved from 11.0 ± 2.4 to 12.2 ± 2.0 (p < 0.001). For the mean evaluation scores of fracture line increased from 12.8 ± 1.7 to 13.5 ± 1.2 (p<0.001) in AP view, and 13.6 ± 1.8 to 14.2 ± 1.5 (p< 0.001) in lateral view. Finally, the mean of soft tissue swelling improved from 12.0 ± 2.1 to 13.11 ± 1.8 (p< 0.001) as shown in **Table 3**.

The multiple regression analysis of association between baseline characteristics and diagnostic scores showed no association as shown in **Table 4**. However, the increasing of age and work experience tend to improve the diagnostic scores with the coefficient 0.25 (-0.66, 1.15), and 0.13 (-0.47, 0.72), respectively.

## 4. Discussion

Our findings demonstrated that pediatric elbow radiographic guidance significantly enhanced the ability to evaluate essential parameters and diagnose lateral condyle fractures. Although

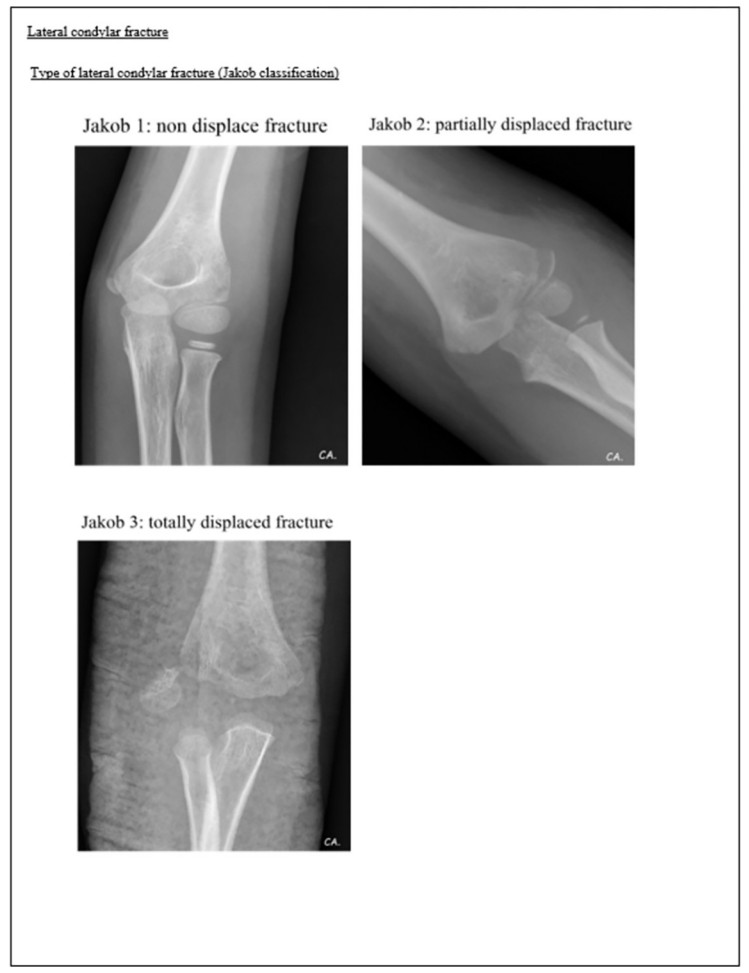

Lateral condylar fracture

Type of lateral condylar fracture (Jakob classification)

Jakob 1: non displace fracture

Jakob 2: partially displaced fracture

Jakob 3: totally displaced fracture

**Fig 5. Pediatric elbow radiographic evaluation guidance (5th page).**

there was no supportive data to indicate what score of improvement should be considered to be clinically significant, we can still pragmatically interpret that one-point improvement of the mean diagnostic score means one more patient accurately diagnosed for proper management

According to a previous study, the prevalence of missed diagnosis of lateral condyle fracture was as high as 60% [7]. In our study, the rate of missed diagnosis was 23% before using the guidance. This may be due to the use of questionnaire for essential parameters evaluation with a specific answer format. The answer format might have potentially guided the participants to evaluate all mandatory parameters and lead to the correct diagnosis, and the missed diagnosis may have been underestimated. Moreover, the final diagnosis was in binary form (yes or no), and participant making a correct diagnosis by chance cannot be ruled out. The study questionnaire could be considered as a limitation of this study reflecting the use of a questionnaire did not resemble a real situation.

In spite of the influential effect of the questionnaire to influence the participants and lower the rate of missed diagnosis in pre-test evaluation, the guidance still showed significant improvement of the diagnostic scores with the missed diagnosis dropping to 16% at post-test compared with 23% at pre-test. Participants' diagnostic scores of fracture line (AP view and lateral view) in Jacob type I improved after the students and residents read the pediatric elbow

---

**Pre-test and post-test questionnaire**

**The effectiveness of the pediatric elbow radiographic guideline**

---

**The questionnaire has 2 parts**

The selection criteria for research participants (must be at least one item).

1. ☐ A fifth or sixth year medical student (Who already passed orthopedics subject)

2. ☐ Resident in radiology department

3. ☐ Resident in emergency medicine department

4. ☐ Resident in Orthopedics

**Do you agree to participate in this survey?**

☐ yes       ☐ No

| Part 1 : Baseline Data | | | Code | |
|---|---|---|---|---|
| Sex | ☐ 1. Female ☐ 2. Male | | sex | [   ] |
| Age | ☐ 1. <25 year ☐ 2. 25-30 year ☐ 3. > 30 year | | age | [   ] |
| Years' experience | ☐ 1. Studying ☐ 2. 1-3 year ☐ 3. > 3 year | | workyear | [   ] |
| How many patient with lateral condyle fracture you have treated? | ☐ 1. < 5 ☐ 2. > 5 | | case | [   ] |
| Orthopedic exam results in year 5 | ☐ 1. D, D+ ☐ 2. C,C+ ☐ 3. B,B+ ☐ 4. A ☐ 5. Unknown | | grade1 | [   ] |
| Orthopedic exam results in year 6 | ☐ 1. D,D+ ☐ 2. C,C+ ☐ 3. B,B+ ☐ 4. A ☐ 5. Unknown | | grade2 | [   ] |
| Have you ever trained by orthopedics resident? | ☐ 1. Yes ☐ 2. No | | fix ward | [   ] |

**Fig 6. The questionnaire part1.**

radiographic guidance. Participants who were miss diagnosed by fracture line AP view in pre-test was 42.18% whereas, at post-test, only 14.50% were misdiagnosed (p-value<0.001). Similarly, diagnosis of participants with fracture line lateral view improved with misdiagnosis at pre-test 16% compared with, only 4.7% misdiagnosed at post-test (p-value = 0.006).

Even after the use of guidance, fractures were still being missed by medical students, residents, and fellows. This points out that, for better diagnostic accuracy in lateral condyle fractures, professional exposure and experience may be more important than only nourishing with explicit knowledge in radiographic evaluation. In order to foster our young health professionals, real case-based teaching, coaching, mentoring and continuing practice should be emphasized in our medical education system.

According to previous literatures, many radiographic parameters are used for assessment of the pediatric elbow fractures, both in coronal and sagittal views. Poppelaars et al demonstrated that angle measurement of posterior fat pad has high inter and intra-observer reliability, with intraclass correlation coefficient of 0.95 (95% CI, 0.91–0.98) and 0.91 (95% CI, 0.41–0.99), respectively. In addition, the author recommended that the anterior fat pad of 16 degrees or

| Part2 : Radiographic assessment | | | |
|---|---|---|---|
| **Case 1** | | | **Code** |
| Posterior fat pad sign | ☐ 1. negative | ☐ 2. positive | pfp1a [ ] |
| Anterior humeral line | ☐ 1. normal | ☐ 2. abnormal | ahl1a [ ] |
| Radiocapitellar line | ☐ 1. normal | ☐ 2. abnormal | fllat1a [ ] |
| Fracture line AP view | ☐ 1. absent | ☐ 2. present | st1a [ ] |
| Fracture line lateral view | ☐ 1. absent | ☐ 2. present | rc1a [ ] |
| Soft tissue swelling | ☐ 1. not swelling | ☐ 2. swelling | flap1a [ ] |
| Lateral condylar fracture | ☐ 1. no | ☐ 2. yes | fx1a [ ] |
| **Case 2** | | | **Code** |
| Posterior fat pad sign | ☐ 1. negative | ☐ 2. positive | pfp2a [ ] |
| Anterior humeral line | ☐ 1. normal | ☐ 2. abnormal | ahl2a [ ] |
| Radiocapitellar line | ☐ 1. normal | ☐ 2. abnormal | fllat2a [ ] |
| Fracture line AP view | ☐ 1. absent | ☐ 2. present | st2a [ ] |
| Fracture line lateral view | ☐ 1. absent | ☐ 2. present | rc2a [ ] |
| Soft tissue swelling | ☐ 1. not swelling | ☐ 2. swelling | flap2a [ ] |
| Lateral condylar fracture | ☐ 1. no | ☐ 2. yes | fx2a [ ] |
| **Case 3** | | | **Code** |
| Posterior fat pad sign | ☐ 1. negative | ☐ 2. positive | pfp3a [ ] |
| Anterior humeral line | ☐ 1. normal | ☐ 2. abnormal | ahl3a [ ] |
| Radiocapitellar line | ☐ 1. normal | ☐ 2. abnormal | fllat3a [ ] |
| Fracture line AP view | ☐ 1. absent | ☐ 2. present | st3a [ ] |
| Fracture line lateral view | ☐ 1. absent | ☐ 2. present | rc3a [ ] |
| Soft tissue swelling | ☐ 1. not swelling | ☐ 2. swelling | flap3a [ ] |
| Lateral condylar fracture | ☐ 1. no | ☐ 2. yes | fx3a [ ] |

**Fig 7. The questionnaire part2.**

more and any visible posterior fat pad are considered positive fat pad signs or abnormal and would help to identified the fractures [17].

Anterior humeral line is also beneficial for evaluation of elbow fractures. Abnormal anterior humeral line on lateral radiographs indicated displaced fractures. Yao B et al described that anterior humeral line has good intraclass correlation coefficient (0.81) only if it is measured on a non-rotated lateral radiograph. Therefore, the quality of the radiographs or the position of the patients during the x-ray processes are important [18].

Radiocapitellar line has been recommended as a radiographic parameter to evaluate the alignment between proximal radius and capitellum. However, using the radiocapitellar line in young children may have some limitation. Fader LM et al described that the radiocapitellar line does not reliably intersect the central third of capitellum in girls age less than 10 years and boys age less than 11 years due to the eccentric ossification center of capitellum [19].

**Table 1. Baseline characteristic.**

| | Medical student (n = 60) (n) | Emergency medical student (n = 32) (n) | Orthopedic resident and fellows (n = 57) (n) | Radiology resiident and fellows (n = 28) (n) |
|---|---|---|---|---|
| | | | n = 177 | |
| Age | | | | |
| < 25 years old | 58 | 0 | 0 | 1 |
| 25–30 years old | 1 | 30 | 34 | 24 |
| >30 years old | 1 | 2 | 23 | 3 |
| Gender | | | | |
| Female | 37 | 21 | 6 | 24 |
| Male | 23 | 11 | 51 | 4 |
| Work experience | | | | |
| Studying | 60 | 3 | 3 | 2 |
| 1–3 years | 0 | 11 | 16 | 4 |
| >3 years | 0 | 18 | 38 | 22 |
| Number of case experience | | | | |
| < 5 cases | 60 | 22 | 34 | 23 |
| > 5 cases | 0 | 10 | 23 | 5 |
| Orthopedic grade in medical school year 5 | | | | |
| Grade A | 4 | 2 | 8 | 1 |
| Grade B, B+ | 25 | 11 | 30 | 6 |
| Grade C, C+ | 4 | 5 | 14 | 3 |
| Grade D,D+ | 1 | 0 | 0 | 0 |
| Not remember | 26 | 14 | 5 | 18 |
| Orthopedic grade in medical school year 6 | | | | |
| Grade A | - | 3 | 16 | 4 |
| Grade B, B+ | - | 13 | 22 | 4 |
| Grade C, C+ | - | 1 | 7 | 1 |
| Grade D,D+ | - | 0 | 0 | 0 |
| Not remember | - | 15 | 12 | 19 |
| Experience in orthopedic internship | | | | |
| Yes | 0 | 0 | 14 | 0 |
| No | 60 | 32 | 43 | 28 |

The strength of our study was the inclusion of various degrees of fracture severity as well as comparative levels of orthopedic knowledge and practical experience among participant groups. The limitations of this study included the fact that the study was conducted in one

**Table 2. Diagnostic score improvement.**

| Group | Diagnostic score (mean(SD)) | | Mean difference (95%CI) | p-value |
|---|---|---|---|---|
| | Pre-test | Post-test | | |
| Medical student | 11.9 (1.9) | 13.1 (1.3) | 1.18 (0.66,1.70) | <0.001* |
| ER | 10.9 (1.8) | 11.8 (2.2) | 0.94 (0.16, 1.71) | <0.001* |
| Orthopedics | 12.9 (1.6) | 13.7 (1.3) | 0.75 (0.30, 1.21) | 0.002* |
| Radiology | 12.5 (1.6) | 12.8 (1.5) | 0.21 (-0.52, 0.96) | 0.558 |
| **Overall** | **12.4 (2.0)** | **13.4 (1.4)** | **1.0 (0.47, 1.52)** | **0.003*** |

*Significant (P<0.05), SD = standard deviation, CI = Confidence interval

**Table 3. Pre-test and post-test of radiographic parameters evaluation score.**

| Radiographic evaluation (N = 177) | Pre-test score, mean (SD) | Post-test score, mean (SD) | Difference (95% CI) | p-value |
|---|---|---|---|---|
| Posterior fat pad sign | 11.21 (2.09) | 12.11 (1.71) | 0.91 (0.57, 1.25) | <0.0001* |
| Medical student | 10.38 (1.72) | 12.32 (1.58) | 1.93 (1.45, 2.42) | <0.0001* |
| ER | 11.03 (2.03) | 11.57 (1.96) | 0.53 (0.47, 1.54) | 0.2860 |
| Orthopedic | 12.03 (2.09) | 12.14 (1.79) | 0.11 (-0.49, 0.70) | 0.7243 |
| Radiology | 11.46 (2.25) | 12.21 (1.47) | 0.75 (0.06, 1.56) | 0.0676 |
| Anterior humeral line | 11.02 (2.37) | 12.22 (2.04) | 1.21 (0.85, 1.57) | <0.0001* |
| Medical student | 10.43 (2.45) | 12.15 (2.07) | 1.72 (1.06, 2.38) | <0.0001* |
| ER | 10.33 (2.20) | 11.53 (2.10) | 1.20 (0.13, 2.27) | 0.0288* |
| Orthopedic | 11.95 (1.95) | 12.89 (1.84) | 0.95 (0.44, 1.46) | 0.0005* |
| Radiology | 11.11 (2.62) | 11.75 (2.01) | 0.64 (0.30, 1.59) | 0.1742 |
| Radiocapitellar line | 12.06 (3.01) | 14.13 (1.86) | 2.07 (1.66, 2.49) | <0.0001* |
| Medical student | 12.03 (3.12) | 14.02 (1.71) | 2.17 (1.39, 2.95) | <0.0001* |
| ER | 9.59 (2.54) | 13.44 (2.10) | 3.86 (2.84, 4.88) | <0.0001* |
| Orthopedic | 13.25 (2.59) | 14.07 (2.04) | 0.82 (0.29, 1.36) | 0.0031* |
| Radiology | 12.25 (2.60) | 14.82 (1.22) | 2.57 (1.53, 3.61) | <0.0001* |
| Fracture line AP | 12.78 (1.69) | 13.49 (1.24) | 0.71 (0.45, 0.97) | <0.0001* |
| Medical student | 12.37 (1.99) | 13.37 (1.35) | 1.00 (0.47, 1.52) | 0.0003* |
| ER | 12.6 (1.71) | 13.03 (0.96) | 0.43 (-0.26, 1.13) | 0.2144 |
| Orthopedic | 13.16 (1.40) | 13.79 (1.25) | 0.63 (0.20, 1.06) | 0.0050* |
| Radiology | 13.11 (1.29) | 13.64 (13.22) | 0.53 (0.11, 0.96) | 0.0162* |
| Fracture line lateral view | 13.66 (1.83) | 14.20 (1.54) | 0.54 (0.30, 0.78) | <0.0001* |
| Medical student | 12.63 (2.04) | 13.38 (1.68) | 0.75 (0.32, 1.17) | 0.0009* |
| ER | 13.5 (1.50) | 14.26 (0.98) | 0.77 (0.16, 1.37) | 0.0144* |
| Orthopedic | 14.34 (1.43) | 14.73 (1.36) | 0.39 (0.05, 0.84) | 0.0858 |
| Radiology | 14.64 (1.16) | 14.78 (1.34) | 0.14 (-0.37, 0.66) | 0.5732 |
| Soft tissue swelling | 12.00 (2.14) | 13.11 (1.81) | 1.11 (0.78, 1.44) | <0.0001* |
| Medical student | 11.08 (2.29) | 12.51 (1.84) | 1.42 (0.81, 2.04) | <0.0001* |
| ER | 11.40 (2.04) | 12.33 (1.75) | 0.93 (0.01, 1.87) | 0.0504 |
| Orthopedic | 12.70 (1.68) | 13.66 (1.63) | 0.96 (0.43, 1.49) | 0.0005* |
| Radiology | 13.18 (1.74) | 14.11 (1.31) | 0.93 (0.09, 1.77) | 0.0310* |

*Significant (P<0.05), SD = standard deviation, CI = Confidence interval

university hospital, therefore, the results may not be applicable to other tiers of hospitals. We only provided AP and lateral radiographs which are standard views in routine practice. However, the oblique radiographs are very important in diagnosing lateral condyle fractures. More

**Table 4. Multiple regression analysis for association between baseline characteristic and diagnostic score improvement.**

| Variables | Coefficient | 95%CI | Standard error | p-value |
|---|---|---|---|---|
| Age | 0.25 | -0.66, 1.15 | 0.45 | 0.591 |
| Work experience (year) | 0.13 | -0.47, 0.72 | 0.30 | 0.668 |
| Case experience ($\geq$ 5 cases) | -0.03 | -0.83, 0.77 | 0.41 | 0.936 |
| 5th year grade $\geq$ B | -0.13 | -1.09, 0.82 | 0.48 | 0.780 |
| 6th year grade $\geq$ B | 0.08 | -0.86, 1.01 | 0.47 | 0.872 |
| Orthopedic internship | 0.32 | -0.89, 1.53 | 0.61 | 0.715 |

*Significant (P<0.05), CI = Confidence interval

studies should be conducting including oblique radiographs. If the oblique radiograph is found to be particularly useful in the diagnosis of lateral condyle fractures, then it should be introduced into routine practice as a standard radiographic view. This study was conducted in 2016 which was many years ago, so it may lag of some updated information. However, the pattern of lateral condyle fracture seems to be the same with unchanged classification during these years. Moreover, the questionnaire was radiographic based and with lack of clinical information, which may alter decision making. A multicentered study with clinical and radiographic information should be done in the future.

## 5. Conclusion

The pediatric elbow radiographic guidance, using PARFS steps, is beneficial for radiographic evaluation and diagnosis, especially for low experienced physicians and trainees. It should be recommended for use in routine medical education and general practice. However, more studies on the impact of pediatric elbow radiographic guidance on students and residents especially with the inclusion of oblique view radiographs will further provide assurances on the benefits of the guidance on the diagnosis of lateral condyle fractures.

## Author Contributions

**Conceptualization:** Satetha Vasaruchapong, Patarawan Woratanarat, Chanika Angsanunsukh.

**Data curation:** Satetha Vasaruchapong, Patarawan Woratanarat, Chanika Angsanunsukh.

**Formal analysis:** Patarawan Woratanarat, Chanika Angsanunsukh.

**Funding acquisition:** Satetha Vasaruchapong, Chanika Angsanunsukh.

**Investigation:** Satetha Vasaruchapong, Chanika Angsanunsukh.

**Methodology:** Satetha Vasaruchapong, Patarawan Woratanarat, Chanika Angsanunsukh.

**Resources:** Patarawan Woratanarat, Supaneewan Jaovisidha, Chanika Angsanunsukh.

**Supervision:** Patarawan Woratanarat, Chanika Angsanunsukh.

**Validation:** Patarawan Woratanarat, Chanika Angsanunsukh.

**Writing – original draft:** Satetha Vasaruchapong, Tanyaporn Patathong, Chanika Angsanunsukh.

**Writing – review & editing:** Patarawan Woratanarat, Tanyaporn Patathong, Thira Woratanarat, Chanika Angsanunsukh.

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
