## [Decision Letter · Decision Letter 0]

9 Nov 2023

PONE-D-23-31280The efficacy of pediatric elbow radiographic guidance in diagnosis of lateral humeral condyle fracturePLOS ONE

Dear Dr. Angsanuntsukh,

Thank you for submitting your manuscript to PLOS ONE. After careful consideration, we feel that it has merit but does not fully meet PLOS ONE’s publication criteria as it currently stands. Therefore, we invite you to submit a revised version of the manuscript that addresses the points raised during the review process.

ACADEMIC EDITOR:Authors are requested to reply all the queries raised by both the reviewers. Please ensure that your decision is justified on PLOS ONE’s publication criteria and not, for example, on novelty or perceived impact.

We look forward to receiving your revised manuscript.

Kind regards,

Priti Chaudhary, M.S.

Academic Editor

PLOS ONE

3. We note that Ffigure(s) S1, S2, S3, S4 and S5 in your submission contain copyrighted images. All PLOS content is published under the Creative Commons Attribution License (CC BY 4.0), which means that the manuscript, images, and Supporting Information files will be freely available online, and any third party is permitted to access, download, copy, distribute, and use these materials in any way, even commercially, with proper attribution. For more information, see our copyright guidelines: http://journals.plos.org/plosone/s/licenses-and-copyright.

a. You may seek permission from the original copyright holder of figure(s) S1, S2, S3, S4 and S5 to publish the content specifically under the CC BY 4.0 license. 

Reviewers' comments:

Reviewer's Responses to Questions

**Comments to the Author**

1. Is the manuscript technically sound, and do the data support the conclusions?

Reviewer #1: Yes

Reviewer #2: Yes

2. Has the statistical analysis been performed appropriately and rigorously? 

Reviewer #1: Yes

Reviewer #2: Yes

3. Have the authors made all data underlying the findings in their manuscript fully available?

Reviewer #1: Yes

Reviewer #2: Yes

4. Is the manuscript presented in an intelligible fashion and written in standard English?

Reviewer #1: Yes

Reviewer #2: Yes

5. Review Comments to the Author

Reviewer #1: introduction addressed the magnitude of the he problem and the aim of the work properly

methods were descried in needed details, validated questionnaires and scientific guidance would have been an addition

results properly addressed with needed statistical data enlisted

discussion configured the week points in a clear manner. references updates is recommended

Reviewer #2: The authors have submitted a manuscript in which they described an additional diagnostic approach in identifying lateral humeral condylar fractures.

Strengths

The authors chose an ongoing medical concern that is related to the scarcity of orthopaedic specialists in Thailand that reflects upon the society in missing one of most common elbow fracture in a vast population of children in a developing country.

Abstract and the introduction were descriptive, proper info regarding the rational, reason and the background for the research were listed, methodology is clear and descriptive. The info listed appears to be sound, the language is clear. Tables and figures are clear and descriptive. No self-citations were found. Oldest citation listed is dating 2001.

Weaknesses

The study was conducted at the interval between 2015-2016, its almost 8 years since the study was conducted. Why did the author/s wait all this time to get their results published.

Line 54: contains a spelling mistake.

Table 1: contains numerical discrepancy between the total numbers listed for med. School students and Orthopaedic grade in medical school year 5 “marked in red”

Discussion is self-focused, no comparison to relevant similar approaches. References are a rather few and can be enriched with the discussion.

6. PLOS authors have the option to publish the peer review history of their article (what does this mean?). If published, this will include your full peer review and any attached files.

Reviewer #1: No

Reviewer #2: No

---

## [Author Response · Author response to Decision Letter 0]

26 Jan 2024

Response to reviewer #1: 

1. Introduction addressed the magnitude of the problem and the aim of the work properly

Answer Thank you so much for your comment.

2. Methods were descried in needed details, validated questionnaires and scientific guidance would have been an addition

Answer Thank you so much for your comment.

3. Results properly addressed with needed statistical data enlisted

Answer Thank you so much for your comment

4. Discussion configured the week points in a clear manner 

Answer Thank you so much for your comment

5. References updates is recommended

Answer Thank you so much for your comment. 

We have updated the recent references in the manuscript as the reviewer recommended. We added the reference number 2, 3, 4, 6, 17, 18, and 19 to the manuscript.

Response to reviewer #2:

The authors have submitted a manuscript in which they described an additional diagnostic approach in identifying lateral humeral condylar fractures.

Strengths

- The authors chose an ongoing medical concern that is related to the scarcity of orthopaedic specialists in Thailand that reflects upon the society in missing one of most common elbow fractures in a vast population of children in a developing country.

- Abstract and the introduction were descriptive, proper info regarding the rational, reason and the background for the research were listed, methodology is clear and descriptive. The info listed appears to be sound, the language is clear. Tables and figures are clear and descriptive. 

- No self-citations were found. Oldest citation listed is dating 2001.

Weaknesses

1. The study was conducted at the interval between 2015-2016, its almost 8 years since the study was conducted. Why did the author/s wait all this time to get their results published?

Answer Thank you so much for your comment. 

We have submitted the manuscript to many journals and has revised it as the reviewers recommended in order to improve the manuscript all along. We also have included this restriction in the discussion as one of limitations of the study, line 210-212. 

“This study was conducted in 2016 which was many years ago, so it may lag of some updated information. However, the pattern of lateral condyle fracture seems to be the same with unchanged classification during these years”

2. Line 54: contains a spelling mistake.

Answer Thank you for your comment. 

We corrected the misspelling in line 54. 

from

 “Moreover, proper position of the elbow and quality of the radiographs are of import”

 to

 “Moreover, the proper position of the elbow and quality of the radiographs are important”

3. Table 1: contains numerical discrepancy between the total numbers listed for med. School students and Orthopaedic grade in medical school year 5 “marked in red”

Answer Thank you for your comment. 

We corrected the number of medical students who cannot remember orthopedic grade in medical school year 5 in Table 1, from 7 to 26. 

4. Discussion is self-focused, no comparison to relevant similar approaches. References are a rather few and can be enriched with the discussion.

Answer Thank you so much for your comments. 

We revised the discussion, added more relevant literatures and updated the references as showed in line 185-201.

5. No visual abstract seen

Answer Thank you so much for your comments. 

We added the visual abstract and submitted with the revised manuscript. 

(File response to reviewers.docx was submitted)

---

## [Editor Report · Decision Letter 1]

8 Feb 2024

The efficacy of pediatric elbow radiographic guidance in diagnosis of lateral humeral condyle fracture

PONE-D-23-31280R1

Dear Dr. Chanika Angsanuntsukh,

We’re pleased to inform you that your manuscript has been judged scientifically suitable for publication and will be formally accepted for publication once it meets all outstanding technical requirements.

Kind regards,

Priti Chaudhary, M.S.

Academic Editor

PLOS ONE
---

## [Editor Report · Acceptance letter]

27 Feb 2024

PONE-D-23-31280R1 

PLOS ONE

Dear Dr. Angsanuntsukh, 

I'm pleased to inform you that your manuscript has been deemed suitable for publication in PLOS ONE. Congratulations! Your manuscript is now being handed over to our production team.

Kind regards, 

on behalf of

Dr. Priti Chaudhary 

Academic Editor

PLOS ONE